# Impact of COVID-19 on mortality in coastal Kenya: a longitudinal open cohort study

M. Otiende [1] ✉, A. Nyaguara[1], C. Bottomley [2], D. Walumbe[1], G. Mochamah[1], D. Amadi[1], C. Nyundo[1], E. W. Kagucia [1], A. O. Etyang[1], I. M. O. Adetifa [1,2], S. P. C. Brand [3], E. Maitha[4], E. Chondo[4], E. Nzomo[5], R. Aman[6], M. Mwangangi[6], P. Amoth[6], K. Kasera[6], W. Ng'ang'a[7], E. Barasa[1], B. Tsofa[1], J. Mwangangi[1], P. Bejon[1,8], A. Agweyu[1], T. N. Williams [1,9] & J. A. G. Scott[1,2]

The mortality impact of COVID-19 in Africa remains controversial because most countries lack vital registration. We analysed excess mortality in Kilifi Health and Demographic Surveillance System, Kenya, using 9 years of baseline data. SARS-CoV-2 seroprevalence studies suggest most adults here were infected before May 2022. During 5 waves of COVID-19 (April 2020-May 2022) an overall excess mortality of 4.8% (95% PI 1.2%, 9.4%) concealed a significant excess (11.6%, 95% PI 5.9%, 18.9%) among older adults ( ≥ 65 years) and a deficit among children aged 1–14 years (−7.7%, 95% PI −20.9%, 6.9%). The excess mortality rate for January 2020-December 2021, age-standardised to the Kenyan population, was 27.4/100,000 person-years (95% CI 23.2-31.6). In Coastal Kenya, excess mortality during the pandemic was substantially lower than in most high-income countries but the significant excess mortality in older adults emphasizes the value of achieving high vaccine coverage in this risk group.

Estimates of global excess mortality during the COVID-19 pandemic in 2020–2021 vary widely from 14.8 million to 19.8 million[1–4]. An important factor driving this variation is uncertainty regarding the impact of COVID-19 in Africa[5]. Across sub-Saharan Africa, most countries lack vital registration systems and only South Africa contributed national mortality data to these models; the estimates of deaths in all other countries were extrapolated from mortality patterns observed elsewhere. In Kenya, national vital registration now records more than half of all deaths but the increasing coverage of deaths in recent years makes it unsuitable to estimate temporal trends in mortality patterns[6]. This lack of data has led to controversy in the interpretation of the pandemic's impact in Africa and this has significant consequences for policy. For example, predictions of a lower

mortality impact in Africa, based on its youthful population structure, stimulated arguments to sustain health spending on existing threats such as malaria, HIV, and respiratory tract infections in children rather than redirect funding to COVID-19 response measures[7]. Health and demographic surveillance systems (HDSS), which conduct longitudinal population-based mortality surveillance, provide an alternative source of mortality data. They cover only a fraction of the national population and cannot be considered representative of the whole country, but they provide a robust empiric insight into the longitudinal mortality experience of selected African populations throughout the pandemic.

In Africa, ascertainment of COVID-19-specific deaths was constrained by limited access to COVID-19 testing. In such settings, excess

[1]KEMRI-Wellcome Research Trust Programme, PO Box 230 Kilifi 80108, Kenya. [2]Department of Infectious Disease Epidemiology, London School of Hygiene & Tropical Medicine, Keppel Street London, London WC1E 7HT, UK. [3]The Zeeman Institute for Systems Biology and Infectious Disease Epidemiology Research, University of Warwick, Coventry CV4 7AL, UK. [4]Department of Health, Kilifi County, Kilifi, Kenya. [5]Kilifi County Hospital, Kilifi, Kenya. [6]Ministry of Health, Government of Kenya; Afya House, Cathedral Road, Nairobi, Kenya. [7]Presidential Policy and Strategy Unit, The Presidency, Government of Kenya, Nairobi, Kenya. [8]Nuffield Department of Clinical Medicine, University of Oxford, Old Road Campus, Oxford OX3 7BN, UK. [9]Institute for Global Health Innovation, Imperial College, London SW72AS, UK. ✉e-mail: motiende@kemri-wellcome.org

mortality is a more useful measure of COVID-19 impact. It sums deaths attributable to COVID-19 and to pandemic restrictions, such as reduced access to health care, and it subtracts mortality gains attributable to pandemic restrictions, such as the suppression of influenza transmission[8–11]. It is calculated as a ratio: the total number of deaths observed in a population divided by the number of deaths expected on the basis of the population mortality experienced in previous years. It can also be presented as a rate: the number of deaths in excess of expectation since the start of the pandemic divided by total person-years of observation.

In the analysis presented here, we estimate both measures of excess mortality in an HDSS population of 306,000 individuals in Kilifi HDSS, Kenya, which has been under continuous surveillance for over 20 years. The same HDSS was also used as a sampling frame for population-based studies of anti-SARS-CoV-2 IgG antibodies. Seroprevalence among adults was 25% in December 2020 to April 2021[12] and 75% from February to May 2022[13]; seroprevalence was lower in children, at 15% and 64%, respectively. Given that only 17% of the adult population had been vaccinated by the time of the second survey, the seroprevalence suggests widespread dissemination of SARS-CoV-2 in Kilifi, typical of elsewhere in Africa[14]. Because excess mortality is dependent on the population age-structure[15] we have reported age-stratified data. We have estimated monthly excess mortality from 1st April 2020 to 5th May 2022, which includes the first five waves of SARS-CoV-2 in Kenya.

## Results

We analysed 16,177 deaths occurring between 1st Jan 2010 and 5th May 2022 from among 3,330,071 PYO. This covered the first 5 waves of COVID-19 in Kenya (Fig. 1). Observed and expected monthly mortality rates are shown in Fig. S1. We found no evidence of autocorrelation (Fig. S2) or lack of fit (Fig. S3) and adjustment for air temperature did not significantly improve our model (Supplement, Fig. S4 and Table S4). We excluded infants from all analyses given the constraints in detecting births accurately during the lockdown period. On aggregating all ages except infants, there was significant excess mortality in November–December 2020, July–August 2021 and December

2021–January 2022 (Fig. 2). These periods coincide with the peak of wave 2 (wild-type), the rise of wave 4 (Delta) and the rise and peak of wave 5 (Omicron BA1), respectively.

We defined the start of a wave as the point at which the daily effective reproduction number of SARS-CoV-2[16,17] traversed 1 in a positive direction after at least 4 weeks below 1 and the end of the wave to be the point where the subsequent wave begins. For wave-specific excess mortality, we offset analysis periods associated with each infection wave by a lag of 2 weeks. There was significant excess mortality for all ages together, excluding infants, during wave 4 (Delta) but not in any other wave (Table 1). We predicted 610 deaths during the Delta wave but observed 711 (excess mortality 16.6%, 95% PI 9.5%, 24.7%). In age-specific analyses, there was significant excess mortality among those aged 5–14 years and 15–44 years in wave 1 (wild-type), among those aged 45–64 years in wave 4 (Delta) and among those aged ≥65 years in waves 4 (Delta) and 5 (Omicron). Mortality was significantly lower than predicted in wave 1 (wild-type) among those aged 45–64 and ≥65 years and in wave 4 (Delta) among those aged 5–14 years. In the three months before the start of the pandemic (1st January–31st March 2020), we observed an overall excess mortality of 14.3% (95% PI 5.1%, 24.9%) which followed a year, 2019, with a significant mortality deficit (Table S3a). In age-specific analyses, this excess mortality in January–March 2020 was significant only among those aged ≥65 years (17.5%, 95% PI 2.9%, 43.9%).

On aggregate, across the first five waves of the pandemic, 1st April 2020–16th April 2022 (Table 2), we predicted 2336 deaths in adults and children aged ≥1 year based on 9 years of baseline data (2010–2018), but we observed 2447 deaths, giving an excess mortality of 4.8% (95% PI 1.2%, 9.4%) and an excess mortality rate of 20.3/100,000 person-years. Excess mortality only deviated significantly above zero among adults aged ≥65 years (11.6% 95% PI 5.9%, 18.9%). Summary excess mortality in all children aged 1–14 years was −7.7% (95% PI −20.9%, 6.9%), but this mortality deficit was mostly driven by improved survival in children aged 5–14 years (Table 2). In sex-specific analyses of the 5-wave period, overall excess mortality was higher in females (9.6%, Table S1a) compared to males (−0.1%, Table S1b). Among females aged 45–64 and ≥65 years, there was a significant excess mortality of 11.4%

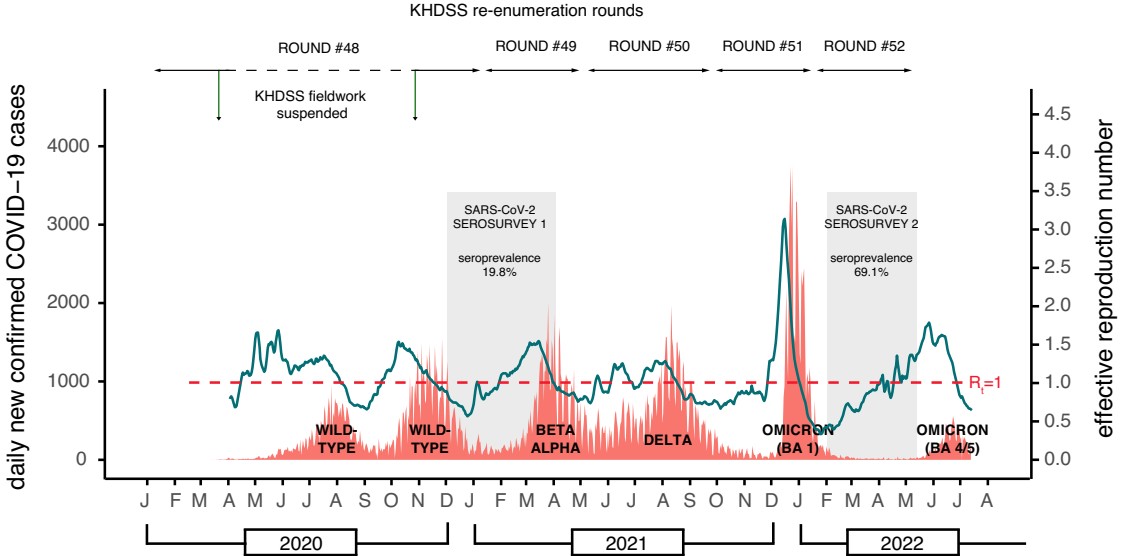

**Fig. 1 | Timeline of the first six COVID-19 waves in Kenya, the Kilifi HDSS re-enumeration rounds and excess mortality analysis windows.** The dotted black horizontal line shows the period when HDSS fieldwork was suspended during round 48. The orange data series is the daily number of new cases of test-positive COVID-19 cases reported in Kenya (scale on left-hand y-axis) [Data source: https://coronavirus.jhu.edu/map.html]. The predominant variant behind each wave is denoted at the base of each wave. The green line represents the effective reproductive number (scale on the right-hand y-axis) from a secondary source[16,17]. The dates of the two anti-SARS-CoV-2 antibody serosurveys in Kilifi HDSS are shown as grey bars[12,13]. The exact dates of the re-enumeration rounds, defined waves and analysis windows are listed in Table S8.

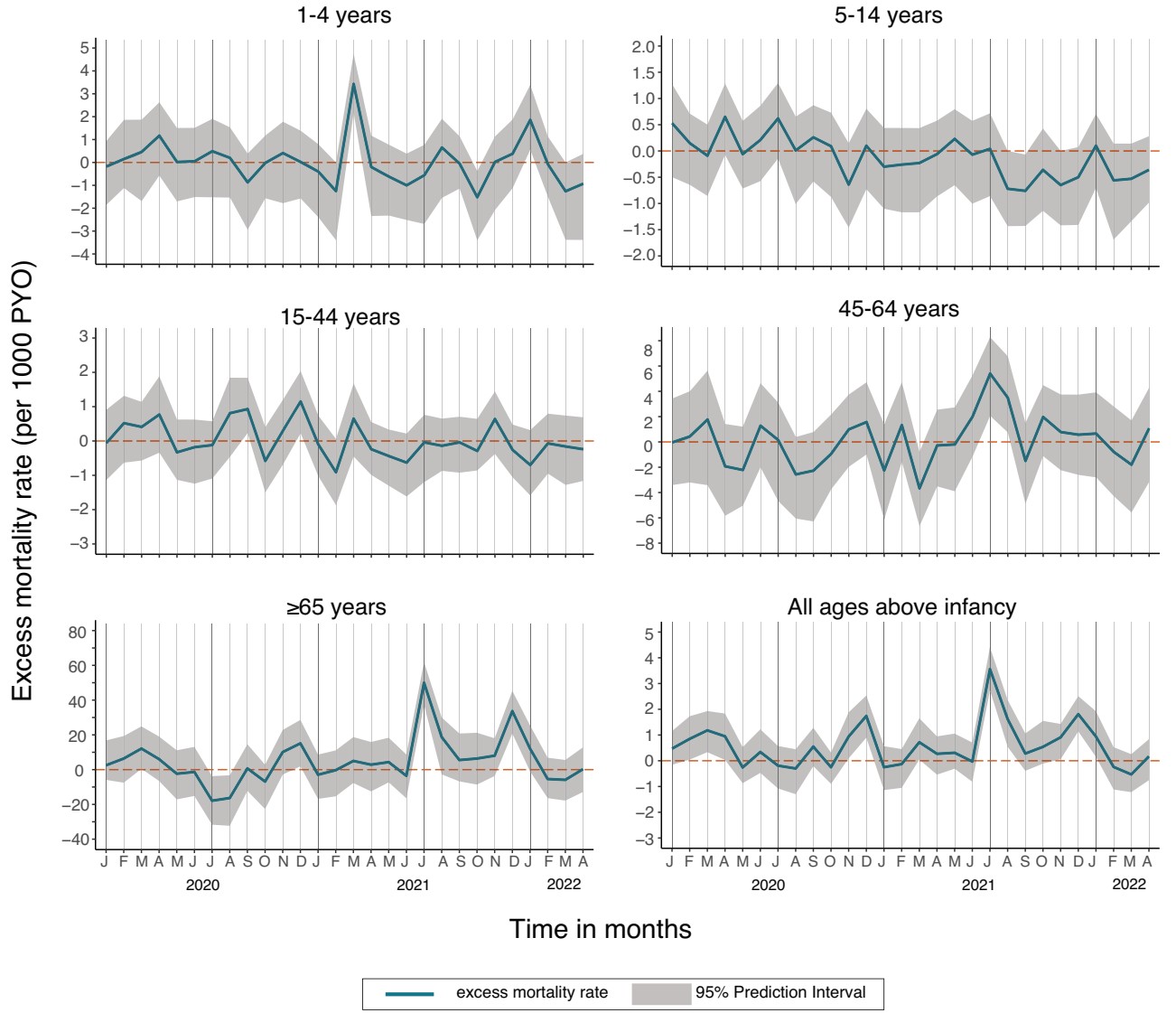

**Fig. 2 | Monthly excess mortality rates from 1st April 2020 to 30th April 2022.** Calculated as (observed deaths−expected deaths)/person-years of observation. The excess mortality rate for all ages above infancy (age ≥1 year) is the weighted average of the age-specific rates. The weights are the proportion of each age group in the Kilifi HDSS population. The grey bands represent the 95% prediction intervals computed as the range from the 5th and 95th percentiles of 100 model simulations.

(95% PI 1.3%, 26.8%) and 16.3% (95% PI 76.4, 27.0), respectively; the equivalent figures for males was −7.6% (95% PI −18.3, 3.5) and 6.3% (95% PI −1.2, 17.3).

We explored the internal validity of the mortality predictions of our baseline model by removing each year (2010–2019) in turn from the analysis and estimating the excess mortality prediction for that year. Predicted mortality differed significantly from observed mortality in 2019 (mortality deficit −7.5%, 95% PI −13.1%, −1.2%, Table S3a). This single deviation is likely to be attributable to random fluctuation in the timing of deaths; it was followed by a brief period of excess mortality in January–March 2020. However, to avoid biasing our baseline model, we excluded data from 2019 when predicting mortality in 2020–2022. Repeating the internal validation analysis for the new baseline (2010–2018) revealed no significant differences between observed and expected deaths (Table S3b).

For comparison with published modelled estimates of global and national excess mortality[1–3] for the two calendar years 2020–2021, we also calculated overall excess mortality rates in Kilifi HDSS for the same period. Observed and expected deaths for all residents aged ≥1 year were 2441 and 2276, respectively, giving an excess mortality of 7.2%

(95% PI 3.4%, 11.3%) and an excess mortality rate of 31.0/100,000 person-years (Table 2). After standardising the Kilifi HDSS results to the age structure of the Kenyan national population, the excess mortality rate was 27.4/100,000 (95%CI, 23.2–31.6) in 2020–2021.

We conducted data quality checks to examine whether bias was introduced as a possible consequence of three factors; (1) field interviewers were unable to reach all household respondents during the pandemic; (2) person-years of observation in the HDSS reduced because delayed fieldwork resulted in a delay in registering new in-migrants; (3) travel restrictions reduced the frequency of migrant labourers returning home to Kilifi for care when they contract a terminal illness[18,19]. If travel restrictions substantially reduced the return of sick diaspora, then the absence of unhealthy in-migrant deaths during the pandemic may have attenuated the excess mortality attributable to COVID-19 among stable residents of Kilifi HDSS. In our analyses: (1) we found no evidence that interviewers were unable to reach household respondents (Table S5); (2) the person-years of observation did decline during the pandemic (Fig. S5), particularly among younger age groups who are, in general, more mobile; however, whilst the risk time for in-migrants was reduced any deaths

**Table 1 | Excess deaths from 1st January 2020 to 16th April 2022 among Kilifi HDSS residents aged ≥1 year, pre-pandemic and in each of 5 COVID-19 waves**

| Age Group | Deaths | | Excess mortality | | | |
|---|---|---|---|---|---|---|
| | Observed | Expected | N | % | 95% PI | Rate/100,000 |
| *1st January 2020–31st March 2020 (pre-pandemic period)* | | | | | | |
| 1–4 years | 15 | 14 | 1 | 7.1 | –33.4, 114.3 | 12.4 |
| 5–14 years | 22 | 18 | 4 | 22.2 | –18.6, 92.1 | 19.0 |
| 15–44 years | 63 | 55 | 8 | 14.5 | –6.7, 41.7 | 30.4 |
| 45–64 years | 71 | 66 | 5 | 7.6 | –14.0, 39.4 | 66.7 |
| ≥65 years | 141 | 120 | 21 | 17.5 | 2.9, 43.9 | 692.7 |
| All ages[a] | 312 | 273 | 39 | 14.3 | 5.1, 24.9 | 59.1 |
| *1st April 2020–4th October 2020 (Wave 1—wild type)* | | | | | | |
| 1–4 years | 31 | 29 | 2 | 6.9 | –18.5, 53.5 | 12.4 |
| 5–14 years | 46 | 35 | 11 | 31.4 | 3.3, 93.2 | 25.8 |
| 15–44 years | 124 | 107 | 17 | 15.9 | 0.3, 38.6 | 31.6 |
| 45–64 years | 111 | 132 | –21 | –15.9 | –26.3, –0.1 | –135.7 |
| ≥65 years | 257 | 294 | –37 | –12.6 | –20.9, –1.9 | –584.1 |
| All ages[a] | 569 | 597 | –28 | –4.7 | –11.0, 4.5 | –20.9 |
| *5th October 2020–14th February 2021 (Wave 2—wild type)* | | | | | | |
| 1–4 years | 17 | 17 | 0 | 0.0 | –36.7, 78.3 | 0.0 |
| 5–14 years | 20 | 27 | –7 | –25.9 | –46.6, 20.1 | –23.4 |
| 15–44 years | 79 | 77 | 2 | 2.6 | –16.6, 27.4 | 5.2 |
| 45–64 years | 88 | 88 | 0 | 0.0 | –16.4, 25.1 | 0.0 |
| ≥65 years | 200 | 185 | 15 | 8.1 | –5.7, 23.4 | 326.8 |
| All ages[a] | 404 | 394 | 10 | 2.5 | –7.8, 11.6 | 10.5 |
| *15th February 2021–4th June 2021 (Wave 3—Beta-Alpha)* | | | | | | |
| 1–4 years | 20 | 16 | 4 | 25 | –16.8, 124.9 | 43.1 |
| 5–14 years | 21 | 20 | 1 | 5 | –32.0, 62.1 | 4.0 |
| 15–44 years | 61 | 66 | –5 | –7.6 | –25.0, 18.7 | –15.2 |
| 45–64 years | 75 | 81 | –6 | –7.4 | –22.9, 15.1 | –64.6 |
| ≥65 years | 179 | 159 | 20 | 12.6 | –3.0, 38.8 | 529.8 |
| All ages[a] | 356 | 342 | 14 | 4.1 | –7.0, 14.1 | 17.4 |
| *5th June 2021–11th December 2021 (Wave 4—Delta)* | | | | | | |
| 1–4 years | 20 | 26 | –6 | –23.1 | –50.7, 28.1 | –36.7 |
| 5–14 years | 19 | 38 | –19 | –50.0 | –65.3, –30.6 | –43.6 |
| 15–44 years | 102 | 110 | –8 | –7.3 | –23.7, 18.6 | –13.9 |
| 45–64 years | 156 | 128 | 28 | 21.9 | 4.4, 43.5 | 169.4 |
| ≥65 years | 414 | 308 | 106 | 34.4 | 19.3, 51.3 | 1545.6 |
| All ages[a] | 711 | 610 | 101 | 16.6 | 9.5, 24.7 | 71.6 |
| *12th December 2021–16th April 2022 (Wave 5—Omicron BA1)* | | | | | | |
| 1–4 years | 19 | 18 | 1 | 5.6 | –27.7, 92.4 | 9.0 |
| 5–14 years | 15 | 25 | –10 | –40.0 | –59.3, -8.5 | –33.7 |
| 15–44 years | 71 | 77 | –6 | –7.8 | –22.5, 17.0 | –15.2 |
| 45–64 years | 95 | 92 | 3 | 3.3 | –11.7, 32.5 | 26.7 |
| ≥65 years | 207 | 179 | 28 | 15.6 | 1.5, 38.4 | 610.2 |

[a]All ages excluding infants <1 year old.

**Table 2 | Excess deaths from 1st January 2020 to 16th April 2022 and for two calendar years (2020–2021) among Kilifi HDSS residents aged >1 year**

| Age Group | Deaths | | Excess mortality | | | |
|---|---|---|---|---|---|---|
| | Observed | Expected | N | % | 95% PI | Rate/100,000 |
| *1st April 2020–16th April 2022 (Waves 1–5)* | | | | | | |
| 1–4 years | 107 | 106 | 1 | 0.9 | –15.1, 27.5 | 1.6 |
| 5–14 years | 121 | 146 | –25 | –17.1 | –31.3, -2.4 | –14.6 |
| 15–44 years | 437 | 437 | 0 | 0.0 | –8.2, 11.0 | 0.0 |
| 45–64 years | 525 | 521 | 4 | 0.8 | –5.9, 10.3 | 6.3 |
| ≥65 years | 1257 | 1126 | 131 | 11.6 | 5.9, 18.9 | 501.1 |
| All ages[a] | 2447 | 2336 | 111 | 4.8 | 1.2, 9.4 | 20.3 |
| *1st January 2020–31st December 2021* | | | | | | |
| 1–4 years | 107 | 105 | 2 | 1.9 | –16.1, 33.9 | 3.2 |
| 5–14 years | 130 | 142 | –12 | –8.5 | –25.3, 10.7 | –7.2 |
| 15–44 years | 442 | 427 | 15 | 3.5 | –4.9, 15.9 | 7.0 |
| 45–64 years | 519 | 508 | 11 | 2.2 | –4.7, 9.4 | 17.8 |
| ≥65 years | 1243 | 1094 | 149 | 13.6 | 6.6, 21.2 | 588.9 |
| All ages[a] | 2441 | 2276 | 165 | 7.2 | 3.4, 11.3 | 31.0 |

[a]All ages excluding infants <1 year old.

the subset of case data derived from Kilifi County. The surveillance data comprised PCR and rapid antigen tests for SARS-CoV-2, including negative results, collated across multiple Kenyan testing laboratories and national reporting mechanisms[20]. From 23rd March 2020 to 5th May 2022 Kilifi accounted for 79,310 (3.2%) of 2,501,682 PCR tests and 6,339 (1.1%) of 581,648 rapid antigen tests nationwide. Superimposing the national and Kilifi-specific epidemic curves (Fig. S7) illustrates that, in Kilifi, wave 1 (wild-type) was absent, wave 3 (Beta-Alpha) arrived late and wave 4 (Delta) arrived early and was more pronounced.

We analysed the cause of death data from 2010 to 2022 to examine whether there were changes in the causes of death in 2020–2022 that could be attributed to the pandemic. Between 2015–2019, we investigated 71.7% (4647/6480) of deaths detected in Kilifi HDSS using verbal autopsy (Fig. S8). In 2020, 2021 and 2022 the proportions we investigated were 70.4% (914/1298), 63.5% (915/1,440) and 47.2% (180/381), respectively. The proportion of deaths attributable to ARI was lower in 2020 in all age groups but returned to pre-COVID-19 levels in 2021 (Fig. 3). The proportion attributable to road traffic accidents (RTA) rose throughout the pandemic, particularly in those aged 15–44 years. The proportion attributable to stroke rose during the pandemic, particularly in those aged 45–64 years.

In 2020, the World Health Organization proposed 6 new COVID-19 questions for verbal autopsies[21]. Of 103 verbal autopsies where at least one COVID-19 question was positive, 31 were attributed to COVID-19 using the COVID-19 Rapid Mortality Surveillance (CRMS) software[22]. This represents 1.8% of 1724 deaths investigated between April 1, 2020, and April 16, 2022. Among the same 103 positive verbal autopsies reviewed by two physicians, 20 were considered probably related to COVID-19; 9 were possibly related; and 74 were unrelated (Fig. S10).

## Discussion

The results of this longstanding mortality surveillance of a population of 306,000 in coastal Kenya indicate a mortality excess of 4.8% throughout the first five waves of the pandemic in Kilifi. This aggregate figure conceals a higher excess mortality of 11.6% among those aged ≥65 years which is offset by a 7.7% reduction in mortality among children aged 1–14 years. Population excess mortality was only significantly positive during one of the five waves of SARS-CoV-2, the Delta wave; this is consistent with observations elsewhere that disease severity was greatest for the Delta variant[23]. The rise in seroprevalence of SARS-CoV-2 antibodies in Kilifi was much greater in 2021 than in

occurring in this unrecorded risk time were also unobserved by the HDSS; (3) we explored bias due to differential mortality by migration status by conducting survival analyses on a fixed cohort of residents selected on 23rd March 2020 and followed for the duration of travel restrictions (7 months) and compared the survival of this cohort to similar cohorts selected on 23rd March of previous years from 2010 to 2019; there was no evidence of decreased survival in 2020 (Table S6 and Fig. S6).

To explain the slight asynchrony between waves of excess mortality in Kilifi HDSS and the national case surveillance data, we analysed

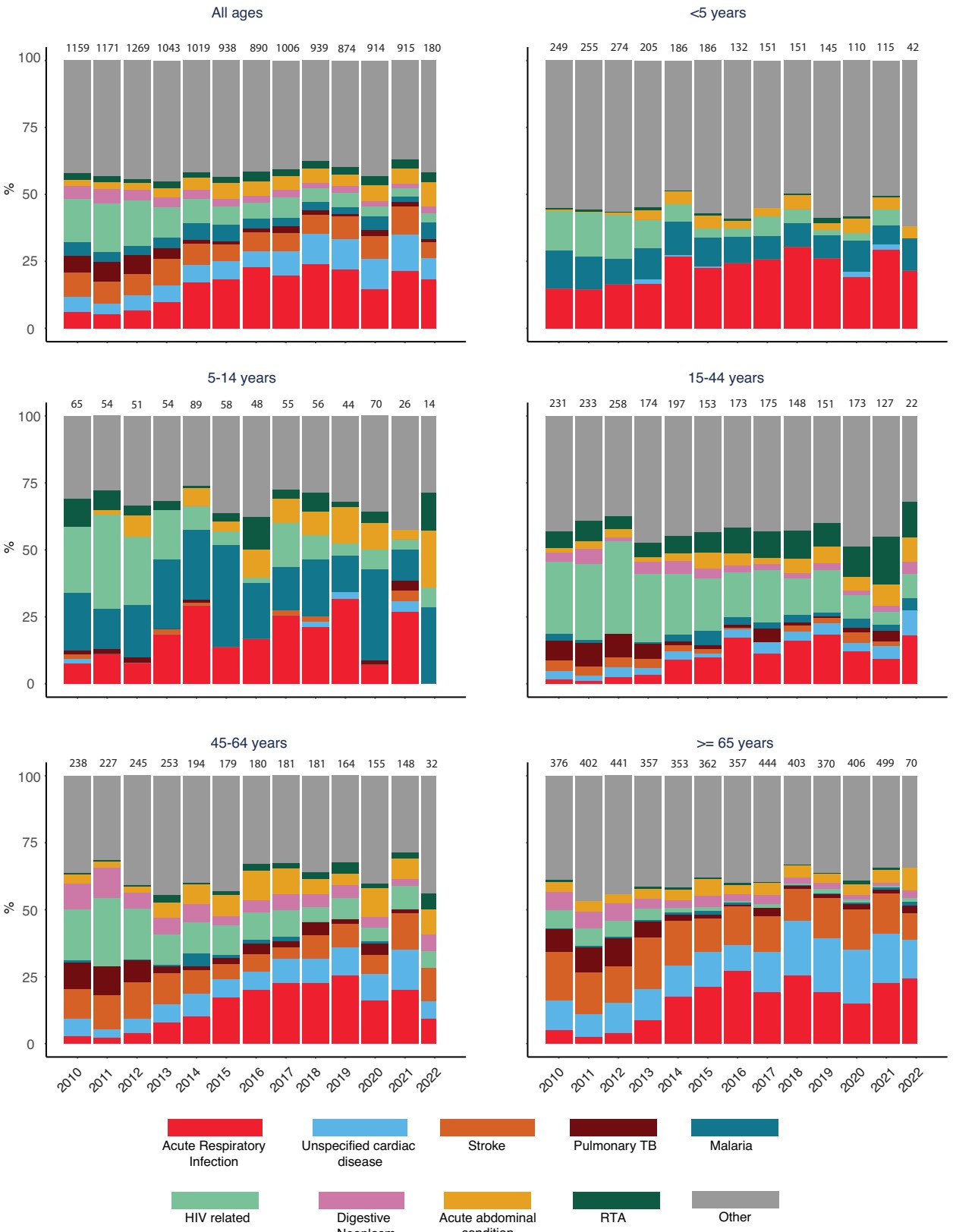

**Fig. 3 | Annual cause-specific mortality fractions by Verbal Autopsy from January 2010 to May 2022.** RTA refers to road traffic accidents.

2020, which may also explain the delayed impact on population mortality.

The strength of our findings is that they are direct empiric observations. Individuals in a large open cohort, who have been followed consistently for 20 years, were enumerated before the pandemic and re-enumerated repeatedly throughout the pandemic. Survival was verified by direct contact with the individual or by the report of a household member or close neighbour. Modelling was used only

to generate a statistical fit to mortality data in the 10-year baseline period to provide a valid prediction of expected mortality during the pandemic period. The model provided a good fit for the baseline data across 9 years and, where it did not, in 2019, we excluded these data to avoid bias. The mortality deficit in 2019 was offset to a substantial degree by a significant mortality excess in the first three months of 2020. Neither of these phenomena could be related to the pandemic as the first case of SARS-CoV-2 infection was detected in Kilifi HDSS on 22nd March 2020. However, in order to compare our empiric results with the predictions of global mortality models we were constrained to begin mortality predictions in January 2020. The stable baseline from 2010 to 2018 provides a robust reference to estimate excess mortality.

The HDSS is, nonetheless, susceptible to observation biases, particularly in March–October 2020 when fieldwork was suspended for 7 months. We may not have been able to ascertain all deaths among infants who were born during this suspension or among older persons who would normally have been expected to migrate into the area. We managed bias attributable to the under-ascertainment of infant deaths by excluding infants from our analyses. We were able to capture fewer PYO during the pandemic, largely because of delays in registering in-migrants into the HDSS, but this was matched by a corresponding loss in the observation of deaths among this unobserved population and this is unlikely to have generated any bias. We explored the possibility that potential in-migrants, grounded elsewhere during the pandemic but intending to return home because of sickness, might have a higher mortality. We examined the potential of in-migrant deaths, absent during lockdown, to obscure a COVID-19 attributable mortality excess among adults, by comparing the mortality experience of the resident cohorts throughout the suspension of fieldwork (23rd March–25th October) in 2020 against the mortality experience of similar cohorts followed through these same dates in the previous 10 years; there was no evidence of attenuation of survival during 2020 in these analyses.

Inferences may be limited by the instability in annual mortality rates in a surveillance population of 306,000. We mitigated temporal variation in the baseline and pandemic periods by fitting our prediction model over 9 years of data (2010–2018) and analysing excess mortality over two years of observation. However, this instability persists in shorter time periods and in subgroup analyses, such as sex-specific analyses. This may explain the excess mortality observed in the three months before the pandemic began, as well as the apparently higher excess mortality in females compared to males. Female excess mortality has been observed to be higher than male excess mortality during the pandemic in some countries[24] but, in Kilifi HDSS, sex-specific estimates of excess mortality were indistinguishable statistically because the 95% prediction intervals were overlapping (Table S1a, S1b).

The inaccessibility of COVID-19 testing in Africa and potential biases in the access to testing have led to controversy regarding the severity of COVID-19 in African populations[25]. Our analysis of excess mortality cannot reliably estimate the infection fatality ratio for SARS-CoV-2 because it cannot distinguish the effects of the pandemic from those of the pandemic response. For example, the negative excess mortality in residents aged 1–14 years in Kilifi suggests there were net mortality benefits attributable to these restrictions; excess mortality could, therefore, substantially underestimate COVID-19-specific mortality in older adults, assuming deaths in young and old are similar and similarly ameliorated by pandemic restrictions. However, models based on seroprevalence data do suggest that the infection fatality ratio in Africa is lower than in Europe and Americas[26,27].

In trying to discriminate these effects we found the VA data had limited value. There was a clear fall in the proportion of deaths due to ARI at all ages in 2020. Restrictions on movement during the pandemic can slow the transmission of existing respiratory pathogens[28]. ARI deaths then reverted to baseline levels in most age groups in 2021, whilst the pandemic waves continued, suggesting the pattern was due

to pandemic restrictions in 2020 and adherence fatigue in 2021. Movement restrictions would also be expected to reduce the proportion of deaths attributable to road traffic accidents but in Kilifi these increased, particularly among the age at highest risk (15–44 years); this paradoxical finding has also been observed in the USA[29].

Among 1724 deaths examined between 1st April 2020 and 16th April 2022, the new WHO VA questions attributed only 31 (1.8%) to COVID-19 suggesting that the COVID-19 questions and algorithms lacked sensitivity in our setting. Deaths from COVID-19 may be mistaken for pneumonia deaths or they may be mediated through thrombotic pathology causing stroke or myocardial infarction[30] and these pathways will not be captured by the COVID-19 VA questions. There is some evidence of an excess in the proportion of stroke deaths during 2021 among middle-aged adults (45–64 y, Fig. 3).

The strongest peak in excess mortality, present at both 45–64 years and ≥65 years, occurred in July 2021. This preceded the peak of the Delta wave in the national surveillance dataset by a few weeks. National serosurveillance and modelling studies have illustrated marked regional heterogeneity in SARS-CoV-2 transmission patterns[20,31] so we examined local COVID-19 testing data from Kilifi County. Although the data are relatively sparse and susceptible to local ascertainment biases, they suggest that wave 4 (Delta) arose earlier in Kilifi by approximately one month (Fig. S7). The local data also help explain the lack of excess mortality during the first wave (wild-type) as there were few infections detected in Kilifi during this wave.

The large negative estimate of excess mortality (−7.7%) among all children aged 1–14 years is a surprising observation which is probably attributable to social restrictions and school closures, limiting the spread of respiratory and gastrointestinal pathogens. Among young children, aged 1–4 years, there is a single peak of excess mortality coincident with the national Beta-Alpha peak but, for the rest of the pandemic, mortality in this age group was lower than predicted. A study from The Gambia, at three HDSSs in Eastern and Western Gambia, identified excess mortality in infants in only one[32].

Whilst excess mortality cannot define the IFR for SARS-CoV-2, it can estimate the net mortality cost to society arising from the total experience of the pandemic. This is important for allocating resources between ongoing COVID-19 control measures and pre-existing health priorities. For Kenya, different global models have yielded widely varying results with profoundly different implications. The Global Burden of Disease models estimated the excess mortality rate in 2020–2021 at 181.2/100,000 whilst the World Health Organization model estimated it at 11/100,000. For comparison, these same models estimated excess mortality rates for the USA at 179.3 and 140/10,000 respectively, and for the UK at 126.8 and 109/100,000, respectively[1–3]. Our estimate of the excess mortality rate in 2020–2021 is 27.4/100,000 after age-standardising Kilifi results to the Kenya population structure. Of note, our estimate excludes the mortality experience of infants though, as noted above, there is little support for an excess of mortality among African children[32,33]. COVID-19 may also be less severe in rural than in urban settings[34] and Kilifi HDSS has more rural residents (~88%) than the national average (69%)[35] so we may be underestimating the national impact slightly. Nonetheless, these empiric data may be generalised to the 69% of the national population that live rural lives and they imply a pandemic mortality impact substantially lower than that observed in the UK or the USA.

Although the measured excess mortality rate associated with the pandemic in coastal Kenya is substantially lower than was predicted in some global models[3,4], it has important implications. Given that 25% of the adults in Kilifi HDSS had no measurable anti-SARS-CoV-2 antibodies when last sampled, a large proportion of the study population remains susceptible to severe or fatal COVID-19 disease. The significant mortality risk identified here, among adults aged ≥65 years in Kilifi, supports a specific focus on the delivery of COVID-19 vaccines to older persons in Kenya.

## Methods

The first case of COVID-19 in Kenya was identified on 12th March 2020 and 323,818 cases and 5649 deaths were reported up 5th May 2022[16]. To date there have been seven waves of COVID-19 in Kenya; the first two waves were driven by the wild-type variant, the third by the Beta/Alpha variants, the fourth by Delta and the remaining three by Omicron (Fig. 1).

Kilifi HDSS was established in 2000[36] with an initial census of 180,000 residents. Vital status and migration events have been recorded in subsequent re-enumeration rounds conducted every 4 months. By 2020, the population size was 306,000 and approximately 1300 deaths were recorded in each of the three years 2017–19[37]. In Kilifi, the first case of COVID-19 was detected on 22nd March 2020. Because of a nationwide lockdown, HDSS field operations were suspended for 7 months between 23rd March 2020 and 25th October 2020.

The cause of death within Kilifi HDSS has been evaluated by Verbal Autopsy (VA) since 2008[38]. Trained field interviewers have questioned close relatives about signs and symptoms of the deceased household member from 30 days after the date of death. The interviews use the 2014 WHO VA questionnaire and are coded by the InterVA-4 algorithm[21]. From April 2020 we asked 6 additional questions (Table S7) that were based on WHO recommendations, intending to identify possible COVID-19 deaths[21].

Mortality rates were calculated as the number of deaths divided by person-years of observation (PYO). PYO was calculated as the time from the latest of birth or in-migration or study start date to the earliest of death or out-migration or study end date. Individuals' periods of residence outside the Kilifi HDSS area were excluded. As excess mortality is temporally linked to COVID-19 waves, models fitted to monthly mortality counts from January 2010 to December 2018 were used to predict a counterfactual scenario for mortality in the absence of COVID-19 from January 2020 to April 2022. We chose January 2010 to December 2018 as the baseline period because mortality rates were stable during this decade[37].

We modelled monthly death counts using negative binomial regression and computed the 95% prediction intervals (95% PI) for each expected death count as the range from the 5th and 95th percentiles of 100 model simulations. The model included a log-linear trend, sine and cosine terms to account for seasonality, and an offset to account for changes in person-years of observation (Equation S1). The model was fitted using the glm.nb function in R. We conducted an internal validation of the model using data from the pre-pandemic period (2010–2018) by excluding one pre-pandemic year at a time and using data from the remaining years to predict mortality for that year. We then assessed the magnitude and direction of the difference between the expected deaths and the observed deaths in the year of interest. Additionally, we compared the distribution of the observed monthly death counts to the predicted distribution from the negative binomial model (Fig. S3). We considered the impact of air temperature on all-cause mortality using air temperature data (for the area covered by the KHDSS) from the Copernicus ERA5 global weather and climate reanalysis dataset[39].

Excess mortality was calculated as the difference between the observed and predicted number of deaths in the COVID-19 period expressed as a percentage of the predicted number of deaths or as a rate per 100,000 person-years observed. We used 1st April 2020 as the start of the analysis period during the pandemic because the interval from infection to death is approximately 2–3 weeks[23] and it is unlikely there would have been any COVID-19 deaths before April 2020. We analysed overall and sex-specific excess mortality from 1st January 2020 to 31st March 2020, in each of the first 5 waves and for the entire duration.

We locked our demographic database on 21st June 2023 and censored our analytic dataset on 5th May 2022, the date our 52nd HDSS re-enumeration round was completed. The interval between censoring and locking allowed ample opportunity to capture deaths that may have been missed during the 52nd round.

The Kilifi HDSS population is re-enumerated every 4 months because infant deaths may be missed with longer intervals; a child may be born and die without any enumeration contact. The suspension of the HDSS field operations on 23rd March 2020 extended this re-enumeration interval from 4 to 11 months and it is likely that the ascertainment of infant deaths was reduced in this period. Therefore, we excluded infants from the estimation of overall excess mortality. In-migrants may also enter and die before they can be enumerated. The increased interval between re-enumeration rounds also reduced the number of PYO, and associated deaths, that were captured among in-migrants. Use of actual PYO to estimate mortality rates, and the correspondence between observed risk time and observed deaths mitigated against any bias in estimating mortality. However, to assess mortality changes that may have arisen due to variable detection of in-migrant deaths, we compared the mortality risk of the snapshot cohort of Kilifi HDSS residents from 23rd March 2020 up to the 25th of October 2020 (the period that fieldwork was suspended) to similar snapshot cohort analyses beginning 23rd March and terminating on 25th October in each year of the baseline period, and in 2021.

Although fieldwork was suspended temporarily in 2020, deaths occurring in the suspension period were ascertained as soon as field studies resumed. However, even after the resumption of field activities, social restrictions may have limited access to some houses, thereby reducing the quality of information gathered; we examined this possibility by analysing the source of each re-enumeration record (household members vs neighbours) during the pandemic years compared to the baseline period.

We calculated cause-specific mortality fractions in 10 categories; the 9 leading causes of death in children and adults and 'all other causes'; the specific causes examined were Acute Respiratory Infections (ARI), unspecified cardiac disease, stroke, pulmonary tuberculosis, malaria, HIV-related deaths, digestive neoplasm, acute abdominal conditions, and Road Traffic Accidents (RTA).

For any death with at least one positive response to the 6 VA COVID-19 questions proposed by the WHO, we applied two discriminating processes. Firstly, we invited two independent reviewers to conduct Physician Certified Verbal Autopsy (PCVA) using clinical information collected during a VA interview and classified each death as a probable, possible or unlikely COVID-19 death. Discordant cases were resolved jointly by the reviewers. Secondly, we used the COVID-19 Rapid Mortality Surveillance (CRMS) software[22], which is a simplified version of the probabilistic modelling methods used in the InterVA-4 models, to derive the probability that the death was COVID-19 related. We used a probability cut-off value of 0.89 based on a validation study conducted in Brazil[40].

Data used in this study were collected and stored in a relational MySQL (version 5.6.46 Enterprise Edition) database using tablets which had electronic forms specified using PHP-MySQL (v7.2.32). All analyses were conducted using STATA/IC version 15.1 (StataCorp College Station, Texas, USA, RRID:SCR_012763) and R version 4.3.1 (RRID:SCR_001905).

### Ethics approval and consent to participate

Individual verbal consent to participate in a continuous health and demographic surveillance system was sought at the household level using a specific informed consent form. Written informed consent was obtained by interviewers from all VA respondents. This study was approved by the Ethical Review Committee of the Kenya Medical Research Institute (approval number: KEMRI/SERU/CGMR-C/007/3057).

### Reporting summary

Further information on research design is available in the Nature Portfolio Reporting Summary linked to this article.

### Data availability

Underlying individual data include geo-located residence and individual hospital records and hence would be a high risk for identifiability. Intermediary data have been published on the Havard Database Server (https://doi.org/10.7910/DVN/HAGRAK) under a CCBY 4.0 license. All data requests will processed by the Data Governance Committee of the KEMRI-Wellcome Trust Research Programme (Email: dgc@kemri-wellcome.org)

### Code availability

Codes for conducting the analysis are also available on the Havard Dataverse Server (https://doi.org/10.7910/DVN/HAGRAK)

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

## Acknowledgements

We gratefully thank the residents of Kilifi who have participated in the surveillance activities of the Kilifi HDSS. We acknowledge the tremendous work of the verbal autopsy and census field staff, data supervisors who collect and process this information, and the Community Liaison Group who run the community engagement programmes. This article is published with the permission of the Director of the Kenya Medical Research Institute. The Wellcome Trust Core support to KEMRI-Wellcome Trust OXF-COR03-2430; Wellcome Trust Research Fellowships 214320 (JAGS) and 202800 (TNW); UK Foreign, Commonwealth and Development Office (SPCB); Wellcome Trust 220985 (SPCB)

## Author contributions

Conceptualisation: J.A.G.S., T.N.W., P.B. Data collection and preparation: A.N., D.W., G.M., D.A., C.N. Formal analysis: M.O., C.B., J.A.G.S. Interpretation: M.O., J.A.G.S., C.B., E.W.K., A.O.E., I.M.O.A., S.P.C.B., E.M., E.C., E.N., R.A., M.M., P.A., K.K., W.N., E.B., B.T., J.M., P.B., A.A., T.N.W. Writing—original draft: M.O., J.A.G.S., C.B. Writing—reviewing & editing: all authors. Resources and funding acquisition: P.B., J.M., B.T., J.A.G.S., T.N.W.

## Competing interests

The authors declare no competing interests.
