## [Peer Review File · Nature Communications]

Impact of COVID-19 on mortality in coastal Kenya: a longitudinal open cohort studyReviewer #1 (Remarks to the Author):

NCOMMS-23-05490-T

Impact of COVID-19 on mortality in coastal Kenya: a longitudinal open cohort study

Summary and comments to the authors

The above-mentioned manuscript examines the excess mortality related with the pandemic in a coastal area in Kenya. Such data and results are important to better understand the impact of the pandemic in developing countries, but the generalisability of this study to Kenya as a whole is questionable and was not discussed in the paper. In addition, I have some considerations about the statistical approach and the lack of a cross validation analysis. I believe the paper also misses structure making hard to read and understand. See below my comments:

- There is no introduction, results and discussion sections making it hard to follow.
- There is not enough literature cited. Part of the "introduction" should have been a paragraph describing efforts for calculating excess. I understand that the literature is massive, nevertheless, a selection of studies are needed here as part of setting the scene.
- It is unclear what the main aim of the study is. This is typically in the last paragraph of the introduction and helps the reader especially in formats like nature communication where the methods section is at the end. In the current stage of the paper, the reader is uncertain about what will read next and the authors have performed more analysis compared to what is suggested in the abstract and couple first paragraphs.
- I do not understand this: "For comparison with modelled estimates of global mortality we also calculated excess mortality in Kilifi HDSS for two calendar years 2020-2021". I thought already the analysis was based on HDSS data and Kilifi. Can you clarify?
- You need to provide more information on how survival analysis tests the existence of under-ascertainment bias introduced because field interviewers were unable to reach households.
- I do not see how the historical cause-specific mortality fractions add to the message of the paper. I would suggest any hypotheses regarding this (and any other additional analysis/bias adjustment) should be stated in the introduction after the aims of the study.

Methodological considerations:

- The pandemic has affected the person years of observation. How did you account for this when fitted the model to predict the counterfactual scenario for mortality in the absence of COVID-19? Have you somehow predicted PYO based on previous years? If no, what did you use and how does this bias your results? The proper thing is to estimate person years had COVID-19 not occurred.
- It is unclear how you accounted for limited access to houses, please clarify.
- The authors need to conduct cross-validation and assess the validity of their model and results. In the cross-validation would be very interesting to see the age-specific bias of your model.
- Some data (I believe that monthly aggregated data at a large region will not be subject to confidentiality considerations) and code should be online available.
- Previous studies have used temperature and national holidays to help predictions, is there any reason for excluding them in your model (given that both covariates are online available, for instance you can retrieve temperature from ERA-5).

Reviewer #2 (Remarks to the Author):

General comments:

- This is an interesting study and the methods appear sound.
- How is the effective reproductive number calculated? This is quite an important part of the paper and some details are needed (unless I missed them).

Specific comments:

- Abstract: I don't think $\%PI$ is standard notation. These are frequentist intervals I believe but please confirm. Do your intervals include negative binomial uncertainty? Or is the uncertainty from parameter uncertainty only?
- Line 43. Check out the Figure 6 in the paper "Estimating global and country-specific excess mortality during the COVID-19 pandemic", accessible at: <https://imstat.org/journals-and-publications/annals-of-applied-statistics/annals-of-applied-statistics-next-issues/>
This figure shows the variation in WHO, Economist, IHME estimates in AFRO, and backs up the claim that there is much uncertainty in African countries.
- Figure 1 has a lot going on and is nice overall but is a little confusing. There are two dashed lines and so the Round 48 comment is ambiguous.
- Can uncertainty be placed on the green line that represents the effective reproductive number in Figure 1?
- Line 89. Define excess mortality since this is not simply ACM-Expected, but I think (ACM-Expected)/Expected which is sometimes called the P-Score. And define what PI is here.
- Line 216. What is the interval estimate on 23.8/100,000?
- Figure S5. Can you explain the statement, "The excess mortality among adults aged ≥ 65 years in the first quarter of 2020 is not accompanied by a change in cause-specific patterns."

Reviewer #1

The above-mentioned manuscript examines the excess mortality related with the pandemic in a coastal area in Kenya. Such data and results are important to better understand the impact of the pandemic in developing countries, but the generalisability of this study to Kenya as a whole is questionable and was not discussed in the paper. In addition, I have some considerations about the statistical approach and the lack of a cross validation analysis. I believe the paper also misses structure making hard to read and understand. See below my comments:

1a. There is no introduction, results and discussion sections making it hard to follow.

We first drafted this paper in conventional style as can be appreciated in our pre-print publication (<https://doi.org/10.1101/2022.10.12.22281019>). We adapted the length and format to the requirements of Nature Medicine as a short report and Nature Medicine referred the paper to Nature Communications without inviting us to reformat. We agree that this shorter, informal, format makes it more taxing to follow. Given this history of the paper, and the advice of the editor to return to a more conventional format, we have redrafted the manuscript and expanded to a conventional length. This also allows us to provide more information on the areas that are identified as missing in several of the reviewers' comments below.

1b. There is not enough literature cited. Part of the "introduction" should have been a paragraph describing efforts for calculating excess. I understand that the literature is massive, nevertheless, a selection of studies is needed here as part of setting the scene.

As noted in 1a. we have now drafted a standard introduction which has cited the relevant literature. The initial version reviewed was subject to a limit of only 30 references and in the redrafted version we have cited 39 references.

1c. It is unclear what the main aim of the study is. This is typically in the last paragraph of the introduction and helps the reader especially in formats like nature communication where the methods section is at the end. In the current stage of the paper, the reader is uncertain about what will read next and the authors have performed more analysis compared to what is suggested in the abstract and couple first paragraphs.

As noted in 1a and 1b, after restructuring the paper the objective of the study is now stated clearly in the last paragraph of the introduction (lines 72-83)

1d. I do not understand this: "For comparison with modelled estimates of global mortality we also calculated excess mortality in Kilifi HDSS for two calendar years 2020-2021". I thought already the analysis was based on HDSS data and Kilifi. Can you clarify?

Clarification: We have examined excess mortality for different periods. Initially we focused on the timing of the pandemic locally and presented our results across the local waves. However, to compare our estimates to those reported by WHO and IHME, we have also calculated Kilifi HDSS excess mortality estimates for the same periods as those selected by the two global modelling studies which are linked to calendar years (i.e., 2020-2021). In making this comparison, we also standardised our mortality estimate to the national age and sex structure. We rephrased this sentence in the paper to make it clear (lines 134-136)

1e. You need to provide more information on how survival analysis tests the existence of under-ascertainment bias introduced because field interviewers were unable to reach households.

We conducted two data quality checks to assess the risk of under ascertainment bias due to; a) field interviewers being unable to reach households, b) travel restrictions during the pandemic impacting migration flows and consequently also impacting mortality among immigrants. The survival analysis

is designed to explore the second of these and does not explore under-ascertainment bias introduced because field interviewers were unable to reach households. In order to examine bias brought about by restrictions of movement (for example people were not able to ‘return home to die’) we conducted 7-month survival analyses on a fixed cohort of residents selected before pandemic restrictions were implemented (23-Mar-2020) and compared the 7-month risk of death in this cohort to similar cohorts selected on 23 March of previous years. We have clarified this in lines 142-156.

If I do not see how the historical cause-specific mortality fractions add to the message of the paper. I would suggest any hypotheses regarding this (and any other additional analysis/bias adjustment) should be stated in the introduction after the aims of the study.

We were keen to understand how causes of death may have changed in 2020-2022 compared to the pre-pandemic period. Looking at trends of cause-specific mortality fractions helped identify changes in acute respiratory illness during the pandemic that were possibly due to pandemic restrictions in 2020 and adherence fatigue in 2021. We have added a sentence in lines 167-169 that justifies why looking at cause of death data was important to us.

Methodological considerations:

1g. The pandemic has affected the person years of observation. How did you account for this when fitted the model to predict the counterfactual scenario for mortality in the absence of COVID-19? Have you somehow predicted PYO based on previous years? If no, what did you use and how does this bias your results? The proper thing is to estimate person years had COVID-19 not occurred.

We used monthly death counts and person years from 2010-2019 to model the counterfactual scenario i.e., expected death counts in 2020-2022 in the absence of the pandemic. Relative excess mortality (p-score) was then calculated as the difference between the observed and predicted number of deaths during the pandemic expressed as a percentage of the predicted number of deaths.

The pandemic affected person years of observation and to calculate excess mortality rates during the pandemic, we estimated the denominator (person years) in the pandemic period by fitting a log-linear model on the observed person years from 2010-2019. We used this model to predict the number of person years in 2020-2022 that would have been expected in the absence of the pandemic. These predicted person years were then used as denominators for rate calculation. In brief, we did estimate person years as if COVID-19 had not occurred.

We have made these details clear in lines 339-346

1h. It is unclear how you accounted for limited access to houses, please clarify.

We suspected that social restrictions may have limited access to some houses and consequently affect the quality of the information collected. Information obtained from a household member is of better quality than information obtained from a neighbour who is not a member of the household. For example, a neighbour may not be aware of a death that has taken place in a large household next door. We conducted an analysis to examine whether social restrictions forced our field staff to rely more frequently on neighbours’ reports during the pandemic, because they could not access the target household (Table S3). This would have reduced the sensitivity of our death detection system during the pandemic. In this analysis, we found no evidence of increased reliance on non-household sources during the pandemic. We have reported this in the methods (lines 373-379)

1i. The authors need to conduct cross-validation and assess the validity of their model and results. In the cross-validation vs thiosould be very interesting to see the age-specific bias of your model. Some data (I believe that monthly aggregated data at a large region will not be subject to confidentiality considerations) and code should be online available.

Now that we have expanded the introduction, we hope it is apparent that there are almost no published data on excess mortality in tropical Africa during the pandemic – the exception being the study covering three HDSS sites in The Gambia which we have made a comparison to in lines 272-274. It is therefore not possible to cross-validate our mortality findings against other comparable settings. It is this absence of data that places such reliance (in public health decision-making) on modelled estimates and part of our purpose is to show the frailty of modelling almost an entire continent in the absence of any specific mortality data from those countries.

To assess the validity of our model we looked autocorrelation plots for each age group and also assessed the fit of the negative binomial distribution when superimposed on the empirical data. We found no evidence of autocorrelation and the negative binomial distribution was a good fit to the empirical distribution of death counts. We have reported this in the methods (lines 335-338) and in the results (lines 91). We have also included illustrations of the autocorrelation plots and the negative binomial distribution fit in the supplementary file.

Finally, we have made our data and code available online at <https://doi.org/10.7910/DVN/HAGRAK>

1k. Previous studies have used temperature and national holidays to help predictions, is there any reason for excluding them in your model (given that both covariates are online available, for instance you can retrieve temperature from ERA-5).

The study spans a duration of over 10 years and there is no consistent source of temperature data locally to include as an adjustment factor. The effect of long-term trends in temperature change will almost certainly be adjusted for in the regression model used to characterise baseline mortality trends over the decade prior to COVID-19, and short-term extremes are (a) uncommon in Kilifi and (b) unlikely to have substantial impact on the majority of our analyses because they consider broad spans of time, e.g. two years (2020-2021).

National holidays affect both the mortality rate and, more especially, the rate of reporting of mortality in high income countries which rely on national vital registration systems. However, the vital registration system in Kenya is insensitive and unreliable and we have not been able to use it. Given the very considerable change in behaviour brought about by the pandemic responses (lockdown – substantially greater than a few national holidays), and given that the annual number of national holidays has not changed between the baseline and pandemic study periods, we believe that the impact of national holidays on mortality will be sufficiently insignificant that we can safely ignore it.

1l. From the preamble. “the generalisability of this study to Kenya as a whole is questionable and was not discussed in the paper”

Part of our purpose is to compare the results of global models of excess mortality with observed excess mortality in order to assess the value of the models. These models are the only current input to guide COVID-19 policy decisions within most African countries. Of note, the models use no mortality data from Kenya whatsoever but do not hesitate to generalise COVID-19 related mortality rates from other continents. We fully accept that the mortality in Kilifi HDSS does not accurately represent the mortality across the whole of Kenya, however, we have attempted to bridge this divide. The principal determinants of excess mortality during the COVID-19 pandemic that vary between Kilifi HDSS and Kenya as a whole are the age structure of the population and the proportion of the population that is urban dwelling. We have adjusted for the first by age-standardizing our excess mortality rates to the age-structure of the Kenya population. We have not adjusted for rural/urban residence but have raised this as a limitation/point of discussion and shared the figures for both settings (Kilifi / Kenya as a whole) which, we believe, will allow readers to assess the (limited) scope for bias in generalising the Kilifi results to compare with modelled estimates of Kenya as a whole.

Reviewer #2

General comments: This is an interesting study and the methods appear sound.

2a. How is the effective reproductive number calculated? This is quite an important part of the paper and some details are needed (unless I missed them).

We did not calculate effective reproductive number for Kilifi HDSS. We used estimates reported for Kenya by another study which we have cited in lines 102-103 and Figure 1

2b. Abstract: I don't think "%PI" is standard notation. These are frequentist intervals I believe but please confirm. Do your intervals include negative binomial uncertainty? Or is the uncertainty from parameter uncertainty only?

We have now specified what the notation 95% PI denotes in the methods and also defined how the intervals were calculated (line 330-332).

The interval quantifies only the stochastic uncertainty i.e., the randomness/variability in the data being modelled using negative binomial regression - not parameter uncertainty.

2c. Line 43. Check out the Figure 6 in the paper "Estimating global and country-specific excess mortality during the COVID-19 pandemic", accessible at: <https://imstat.org/journals-and-publications/annals-of-applied-statistics/annals-of-applied-statistics-next-issues/> This figure shows the variation in WHO, Economist, IHME estimates in AFRO, and backs up the claim that there is much uncertainty in African countries.

Thank you for drawing this paper to our attention which we have now included as support to the claim that there is much uncertainty in the modelling of African country mortality during the pandemic.

2d. Figure 1 has a lot going on and is nice overall but is a little confusing. There are two dashed lines and so the Round 48 comment is ambiguous.

In the caption in figure 1, we have now specified which dashed line we are referring to.

2e. Can uncertainty be placed on the green line that represents the effective reproductive number in Figure 1?

As mentioned in 2a, we used a secondary data source that reported the effective reproduction number for Kenya – but unfortunately without data on uncertainty. We have cited the source accordingly.

2f. Line 89. Define excess mortality since this is not simply ACM-Expected, but I think (ACM-Expected)/Expected which is sometimes called the P-Score. And define what PI is here.

We have defined excess mortality in the introduction (lines 67-71) and in the methods (lines 339-341)

2g. Line 216. What is the interval estimate on 23.8/100,000?

We have calculated the 95% confidence interval for this estimate and reported it in the paper. Age-standardized excess mortality rate 23.8/100,000 (95% CI 20.0 - 27.6).

2h. Figure S5. Can you explain the statement, "The excess mortality among adults aged ≥ 65 years in the first quarter of 2020 is not accompanied by a change in cause-specific patterns."

One of the surprising results of this analysis is the marked excess of mortality in older person (≥ 65 years) in the first three months of 2020, before the pandemic began. We do not have an explanation for this phenomenon (other than random fluctuation). If this was due to a specific cause, we might have expected to see a change in the pattern of cause of death (using verbal autopsy) during that quarter, when compared to preceding years (Figure 3) or when compared to other quarters in 2020

(now Figure S6) but we do not see any such change. When we reduced our paper to a short report format we excised this statement from its location commenting on the excess mortality in the results and put it in the legend of Figure S6 to help justify why the figure was included in the supplement. Now that we have reverted to a conventional paper format we have relocated it in the results text where we hope its context will enable readers to understand our intent (lines 179-180)

REVIEWER COMMENTS

Reviewer #1 (Remarks to the Author):

I would like to thank the authors for addressing some of my comments. Nevertheless, I would like to stress that for a couple of them the answers were inadequate.

I have considerations in the methodology used. This is also highlighted in the results of the paper, where the authors report significant positive excess in the pre-pandemic period:

- I have suggested the inclusion of covariates such as the temperature, which I would like to highlight that it is available online through the ERA5 reanalysis dataset

(<https://cds.climate.copernicus.eu/cdsapp#!/dataset/reanalysis-era5-single-levels?tab=overview>). Previous studies have also modelled the spatial autocorrelation to help predictions (see <https://journals.plos.org/plosone/article?id=10.1371/journal.pone.0240286>).

- This discussion could have been avoided if the authors have provided a cross-validation. When I refer to cross-validation I do not mean comparison with other studies, but rather an internal cross-validation. For example, they can use historical data from 2010-2018, and fit their model and predict the year 2019, and then compare the truth of the year 2019 with the predicted through the model. This will add some validity to the model. (see here for an example of cross validation <https://arxiv.org/pdf/2201.06458.pdf>)

- Is there any justification for the selected age groups? Most of the studies have used different age groups, mostly grouping children and young adults due to the small numbers (see for example <https://www.ncbi.nlm.nih.gov/pmc/articles/PMC8213785/>). It is obvious that this study confronts issues due to the small numbers in these groups, so I am wondering if there is any concrete hypotheses about the selection of these age groups. If there is not, I would suggest some coarser grouping to take care of the small numbers

- The temporal dimension is monthly. Is this because of data availability? Is there information about weekly deaths? If a coarser age grouping is used, I think you should consider a higher resolution in the temporal dimension, say weekly. This will provide a better picture of the temporal trends of the excess in Kilifi.

- You mention in the point by point reply that you have also developed a model to predict person years had COVID-19 not occurred. I could not find this model in the manuscript. Also having developed a model for the person years, you need to provide some validation of the estimates (say similar cross validation as the one suggested above). You also mention something about prediction intervals in the main manuscript. I am wondering if these refer to the sampling of the PY based on this model and the subsequent propagation of this uncertainty in the model for predicting deaths. If not this uncertainty should be propagated, see <https://pubmed.ncbi.nlm.nih.gov/36609356/>.

Reviewer #1

I would like to thank the authors for addressing some of my comments. Nevertheless, I would like to stress that for a couple of them the answers were inadequate.

I have considerations in the methodology used. This is also highlighted in the results of the paper, where the authors report significant positive excess in the pre-pandemic period:

1. I have suggested the inclusion of covariates such as the temperature, which I would like to highlight that it is available online through the ERA5 reanalysis dataset

(<https://cds.climate.copernicus.eu/cdsapp#!/dataset/reanalysis-era5-single-levels?tab=overview>).

Previous studies have also modelled the spatial autocorrelation to help predictions

(see <https://journals.plos.org/plosone/article?id=10.1371/journal.pone.0240286>).

Whilst extremes of temperature, especially summer heat, are a significant driver of excess mortality in Europe and North America, Kilifi benefits from a relatively stable temperature throughout the year being on the equator and close to the Indian Ocean. We have looked at the temperature tables from the ERA5 reanalysis dataset (which are modelled estimates) and we have observed a very narrow range of temperature variation over time in the geographic area covered by the KHDSS which is unlikely to impact on mortality or to confound our analyses. We have confirmed this by fitting a model with daily temperature as a covariate and compared this with the original model. Both models fit the data equally well and we have therefore retained the simpler model. We have also calculated excess mortality estimates based on a model that includes temperature as a covariate. These estimates do not show any significant deviation from the estimates based on a model that does not include temperature. Kilifi HDSS covers only 900 square kilometres and experiences a uniform climate. We therefore do not expect any spatial variation in temperatures. We have made these points in the paper (line 90; 361-363), in the first page of the supplementary file, and included a summary table (Table S4) and a time series of temperature data in the supplementary file (Figure S4).

2. This discussion could have been avoided if the authors have provided a cross-validation. When I refer to cross-validation I do not mean comparison with other studies, but rather an internal cross-validation. For example, they can use historical data from 2010-2018, and fit their model and predict the year 2019, and then compare the truth of the year 2019 with the predicted through the model. This will add some validity in the model. (see here for an example of cross validation <https://arxiv.org/pdf/2201.06458.pdf>)

As suggested by the reviewer, we conducted a cross-validation in the pre-pandemic period (2010-2019) by leaving out one historical year at a time and calculating excess mortality for the year left out. We have done this for each of the baseline years. From this exercise, we found that 2019 was an anomalous year with a significant deficit in all-cause mortality. We know from our existing analysis that this was followed by a significant and unexplained excess mortality in the first 3 months of 2020. We hypothesise that these changes, which are largely balanced, represent a chance variation in the timing of mortality; effectively, this suggests that the excess mortality in early 2020 represents delayed mortality that we would normally have expected to take place in 2019. This anomaly is not related in any way to the pandemic because it occurred before SARS-CoV-2 was established in our area. We considered moving the units of annual baseline observations from January-December to April-March so that the last year in the baseline period would span April 2019 - March 2020, but one

of the key output of our analysis is the comparison of the excess mortality in Jan-2020 to Dec-2021, which is consistent with global models of excess mortality. Because of the potential for bias from this anomalous year, we have dropped 2019 from the baseline estimation, as the period 2010-2018 provided a more robust and consistent baseline from which to generate predictions throughout the pandemic period. We have explained this in the paper in lines 129-138, 210-218

3. Is there any justification for the selected age groups? Most of the studies have used different age groups, mostly grouping children and young adults due to the small numbers (see for example <https://www.ncbi.nlm.nih.gov/pmc/articles/PMC8213785/>). It is obvious that this study confronts issues due to the small numbers in these groups, so I am wondering if there is any concrete hypotheses about the selection of these age groups. If there is not, I would suggest some coarse grouping to take care of the small numbers.

Response

We have used 5 age groups. We had started with 7 but dropped 2 (neonates and infants) because of data quality concerns. There were good reasons to study young children as there was an early concern that young children may be affected by COVID-19¹. Among adults we have distinguished young, middle aged and elderly. This is because COVID-19 has a very strong age gradient, and we would expect to see (and indeed do see) a gradation of effects. The age group 5-14 could be consolidated with young adults, and this may make sense in analyses of industrialised countries. However, in Kilifi they are not an especially small group - approximately 30% of the whole population lies in this group and they have more 'expected deaths' than young children. They also die from a different pattern of causes - as is evident in the VA analysis (Figure 3). The population under surveillance is small (310,000) but because this is an area with high background mortality, we have sufficient events in each of our 5 age groups to detect reasonable deviations - and the deviations brought about by COVID, when they occurred, were large.

4. The temporal dimension is monthly. Is this because of data availability? Is there information about weekly deaths? If a coarser age grouping is used, I think you should consider a higher resolution in the temporal dimension, say weekly. This will provide a better picture of the temporal trends of the excess in Kilifi.

Response:

We have selected 5 age groups and monthly reporting but an equivalent distribution of the data for the same power might be weekly reporting of a single age group. We preferred to evaluate age-related differences in relation to COVID-19, rather than analysing week-to-week variations because (a) each COVID-19 wave lasted several months and (b) the mortality reporting was by recall (not by death registration) and therefore the accuracy of the dates of the deaths do not justify a temporal specification that is too highly resolved.

5. You mention in the point-by-point reply that you have also developed a model to predict person years had COVID-19 not occurred. I could not find this model in the manuscript. Also having developed a model for the person years, you need to provide some validation of the estimates (say similar cross validation as the one suggested above). You also mention something about prediction intervals in the main manuscript. I am wondering if these refer to the sampling of the PY based on this

model and the subsequent propagation of this uncertainty in the model for predicting deaths. If not this uncertainty should be propagated, see <https://pubmed.ncbi.nlm.nih.gov/36609356/>.

Response

We did originally fit a model to the baseline PYO to predict the PYO during the pandemic. However, following this comment above we have reviewed this approach. Originally, we were concerned that the pandemic restrictions leading to disruption of KHDSS field activities might lead us to underestimate the number of person years in the years 2020 onwards and that the PYO would be better predicted by an extrapolation from the modelled baseline. Upon further checking (Figure S5) we found that the pandemic had complex impacts on the observed PYO depending on age. For example, PYOs in infants were particularly affected by the disruption whilst PYOs in adults aged 45+ years were scarcely affected. On considering the problem, we realised that the loss of PYO during the pandemic was also matched by a loss in the observation of corresponding deaths occurring during these lost PYO. The modelled over-estimation of PYO, in our last version, explained to a large degree the mortality deficit that we had calculated in children, because we were observing fewer deaths but assuming an uninterrupted extension of PYO. We therefore revised the analysis and chose not to model PYOs but to include them as directly observed data.

Although changes in field work patterns, brought about by pandemic restrictions, will have had a corresponding impact on PYO and on the matched deaths arising in those PYO, there is still an opportunity for bias if the PYO excluded are of higher-than-average risk. However, we have already considered this bias and analysed it (e.g. unhealthy migrant bias) in our initial submission.

In this version (lines 157-160) we have illustrated the changes in PYO (Figure S5) and commented on these within the data quality section (page 3) in the Supplement. We have now nested the discussion of the unhealthy migrant bias underneath the discussion of this broader question (supplementary file, page 4).

Reference

- 1 Prieto J *et al.* Under- Five Mortality during the Covid-19 Outbreak: Evidence from Four Demographic Surveillance Systems in Low-Income Countries. European Population Conference - EPC 2022, Jun 2022, Groningen, Netherlands. hal-03907638.

REVIEWERS' COMMENTS

Reviewer #1 (Remarks to the Author):

The authors have addressed my comments. I am still worried about the validity of the results:

- Year 2019 was removed. Have you ran an additional cross validation assessing the predictions removing this year?
- Person years are now not used in the analysis and neglecting denominator trends leads to spurious excess. Maybe its worth showing results with and without PY and discuss about the differences.

Reviewer #1

The authors have addressed my comments. I am still worried about the validity of the results:

1. Year 2019 was removed. Have you ran an additional cross validation assessing the predictions removing this year?

Response:

We conducted a cross-validation in the new baseline period (2010-2018) by leaving out one historical year at a time and calculating excess mortality for the year left out. This analysis is presented in supplementary table 3b. We have modified the caption of the table to make this clear.

2. Person years are now not used in the analysis and neglecting denominator trends leads to spurious excess. Maybe its worth showing results with and without PY and discuss about the differences.

Response:

Person years are still used in the analysis. As explained in our previous response, we had initially fit a model to the baseline PYO to predict the PYO during the pandemic. But following a previous comment from the reviewer we have reviewed this approach. Originally, we were concerned that the pandemic restrictions leading to disruption of KHDSS field activities might lead us to underestimate the number of person years in the years 2020 onwards and that the PYO would be better predicted by an extrapolation from the modelled baseline. Upon further checking (Figure S5) we found that the pandemic had complex impacts on the observed PYO depending on age. For example, PYOs in infants were particularly affected by the disruption whilst PYOs in adults aged 45+ years were scarcely affected. On considering the problem, we realised that the loss of PYO during the pandemic was also matched by a loss in the observation of corresponding deaths occurring during these lost PYO. The modelled over-estimation of PYO, in our last version, explained to a large degree the mortality deficit that we had calculated in children, because we were observing fewer deaths but assuming an uninterrupted extension of PYO. **We therefore revised the analysis and chose not to model PYOs but to use directly observed PYO.**

Although changes in field work patterns, brought about by pandemic restrictions, will have had a corresponding impact on PYO and on the matched deaths arising in those PYO, there is still an opportunity for bias if the PYO excluded are of higher-than-average risk. However, we have already considered this bias and analysed it (e.g. unhealthy migrant bias) in our initial submission.

In this version (lines 157-160) we have illustrated the changes in PYO (Figure S5) and commented on these within the data quality section (page 3) in the Supplement. We have now nested the discussion

of the unhealthy migrant bias underneath the discussion of this broader question (supplementary file, page 4).